# Changes in Gut Microbiota Induced by Doxycycline Influence in Vascular Function and Development of Hypertension in DOCA-Salt Rats

**DOI:** 10.3390/nu13092971

**Published:** 2021-08-26

**Authors:** Iñaki Robles-Vera, Néstor de la Visitación, Marta Toral, Manuel Sánchez, Miguel Romero, Manuel Gómez-Guzmán, Félix Vargas, Juan Duarte, Rosario Jiménez

**Affiliations:** 1Department of Pharmacology, School of Pharmacy and Center for Biomedical Research (CIBM), University of Granada, 18071 Granada, Spain; inaki.robles@cnic.es (I.R.-V.); nestorvp@correo.ugr.es (N.d.l.V.); manuelsanchezsantos@ugr.es (M.S.); mgguzman@ugr.es (M.G.-G.); jmduarte@ugr.es (J.D.); rjmoleon@ugr.es (R.J.); 2Gene Regulation in Cardiovascular Remodeling and Inflammation Group, Centro Nacional de Investigaciones Cardiovasculares (CNIC), 28029 Madrid, Spain; marta.toral@cnic.es; 3CIBERCV, 28029 Madrid, Spain; 4Instituto de Investigación Biosanitaria de Granada, 18012 Granada, Spain; fvargas@ugr.es; 5Department of Physiology, School of Medicine, University of Granada, 18071 Granada, Spain

**Keywords:** doxycycline, gut dysbiosis, hypertension, oxidative stress, inflammation, DOCA-salt model

## Abstract

Previous experiments in animals and humans show that shifts in microbiota and its metabolites are linked to hypertension. The present study investigates whether doxycycline (DOX, a broad-spectrum tetracycline antibiotic) improves dysbiosis, prevent cardiovascular pathology and attenuate hypertension in deoxycorticosterone acetate (DOCA)-salt rats, a renin-independent model of hypertension. Male Wistar rats were randomly assigned to three groups: control, DOCA-salt hypertensive rats, DOCA-salt treated with DOX for 4 weeks. DOX decreased systolic blood pressure, improving endothelial dysfunction and reducing aortic oxidative stress and inflammation. DOX decreased lactate-producing bacterial population and plasma lactate levels, improved gut barrier integrity, normalized endotoxemia, plasma noradrenaline levels and restored the Treg content in aorta. These data demonstrate that DOX through direct effects on gut microbiota and its non-microbial effects (anti-inflammatory and immunomodulatory) reduces endothelial dysfunction and the increase in blood pressure in this low-renin form of hypertension.

## 1. Introduction

The use of antibiotics is one of the most important and efficient way of protecting human beings from infections and has increased longevity of life. However, such a rise in lifespan is accompanied by an increase in cardiovascular diseases including stroke and myocardial infarction [1]. Hypertension, the main risk factor for cardiovascular diseases, is due to combination of host genetic, dietary and environmental factors to the chronic increase in blood pressure [2]. Recent studies conducted have demonstrated that, besides the host genome, the collective genomes of the microbiota are important, but largely unknown, as determinants of blood pressure. Several studies have shown that an imbalance in gut microbiota, commonly known as dysbiosis, is associated with high blood pressure in both animal and human hypertension [3,4]. In fact, compared with their respective normotensive strains and models, all experimental models of hypertension studied to date, including the spontaneously hypertensive rat (SHR), Dahl salt-sensitive, angiotensin II (Ang II) and deoxycorticosterone acetate (DOCA)-salt, have altered gut microbiota [3,5,6,7]. In addition, animals lacking gut microbiota have shown to be protected against AngII-induced arterial hypertension, vascular dysfunction and hypertension-induced end-organ damage, demonstrating that commensal microbiota could be an environmental factor inducing Ang II-induced high blood pressure [8]. Therefore, modulation of the gut microbiome through antibiotic treatment may have an effect on the prevalence and origin of disease. Indeed, the non-absorbable antibiotics vancomycin (glycopeptide with antimicrobial activity against Gram-positive organisms) or neomycin (aminoglycoside with wide spectrum of antibacterial action) attenuated steroid-induced hypertension in rats, a model of salt resistant hypertension [9,10]. In addition, the oral neomycin treatment decreased the development of high blood pressure in the young SHR, all of them linking the gut microbiota to hypertension. Furthermore, a recent study has shown that vancomycin and minocycline (a broad-spectrum tetracycline antibiotic) reduce systolic blood pressure (SBP) in older SHR rats with established hypertension [11]. Furthermore, Yang et al., also demonstrated that minocycline, a small molecule that freely passes through the blood–brain barrier, inhibiting microglial activation and rebalance the gut microbiota attenuating the high blood pressure induced by chronic Ang II infusion [3]. Indeed, in this study, minocycline restored the B population, reshaping the microbiota composition by restoring the Firmicutes/Bacteroidetes (F/B) ratio and increasing microbial diversity.

Doxycycline (DOX) belongs to the same antibiotic class than minocycline, indicated by several experimental studies demonstrating that DOX attenuates the increase in SBP in SHR [12], renovascular hypertension [13], lead-induced hypertension [14] and in pregnancy hypertension [15] in rats. Furthermore, in renovascular hypertension DOX treatment also improved arterial wall hypertrophy, excessive collagen/elastin deposition and ameliorated endothelial-dependent vasorelaxion by attenuating oxidative stress and improving nitric oxide (NO) bioavailability [13,16,17]. All these findings showed that DOX produces beneficial effects on renin-angiotensin-dependent rodent models, implicating intimate involvement of gut microbiota in hypertension. A significant number of patients, particularly African Americans, however, exhibits low efficacy for inhibitors of the renin-angiotensin pathway due to low renin profile. Therefore, the therapeutic implications of this potentially novel strategy for control of hypertension remain limited. Thus, the objective of this study is aimed at investigating whether DOX prevents cardiovascular pathology and attenuates hypertension in DOCA-salt rats, a renin-independent model of hypertension. This DOCA model is a neurogenic type of hypertension mostly characterized by elevated sympathetic nerve discharge and altered baroreceptor reflex preceding the rise in blood pressure [18]. Taking account that DOX is not able to adequately penetrate the blood–brain barrier, as compared to minocycline, we can observe blood pressure effects independent of direct action in central nervous system.

## 2. Materials and Methods

### 2.1. Animals and Experimental Groups

The experimental protocol was carried out in compliance with the European Union guide for care and use of laboratory animals. The experimental procedure has been reviewed by the Animal Ethic Committee of Laboratory Animals of the University of Granada, Spain (permit no: 03-CEEA-OH-2013). Male Wistar rats (Envigo Laboratories, Barcelona, Spain), aged 6–7 weeks and initially weighting 150–180 g, were used in this study and housed in a constant temperature (24 ± 1 °C), with a 12 h dark/light cycle. The experimental rats were maintained on standard rat chow (SAFE A04, Augy, France) and tap water ad libitum. The animals used were taken for a period of 2 weeks to adapt for vehicle administration and SBP measurements before the initiation of experimental procedures.

The rats were randomly divided into three experimental groups (*n* = 8): (I) control, (II) DOCA-salt (1% NaCl in drinking water) and (III) DOX-treated DOCA-salt (1% NaCl in drinking water and DOX 25 mg kg^−1^ by oral gavage mixed in 1 mL of 1% methylcellulose, once daily). DOCA-salt hypertension model was induced in uninephrectomized animals by intramuscular injection of DOCA 12.5 mg/0.5 mL/rat per week for four consecutive weeks as previously is described in detail [19]. These animals were maintained in a specific pathogen-free environment at University of Granada Biological Services Unit, in individual ventilated cages. The treatment in total lasted 4 weeks. Weekly body weight, water and food consumption were measured for all experimental groups.

### 2.2. Blood Pressure Measurements

SBP was measured once a week on conscious, pre-warmed and restrained rats by tail-cuff plethysmography (digital pressure meter LE 5000; Letica S.A., Barcelona, Spain) as previously has been described by Gómez-Guzman et al., 2015 [20]. At least ten measurements were made for each rat followed by calculation of the mean of the lowest three values within 5 mm Hg as the SBP level.

### 2.3. Cardiac and Renal Weight Indices

Once the experimental period reached its endpoint, 18 h fasting-animals were anesthetized with 2.5 mL kg^−1^ equitensin (intraperitoneal) and their blood from the abdominal aorta was collected. Finally, the animals were sacrificed by exsanguination. The colon, hearts and kidneys then were removed, cleaned and weighed. The atria and the right ventricle were collected and the remaining left ventricle was weighed. The cardiac, left-ventricular and renal hypertrophic indices were calculated as the heart, left ventricle and kidney weight to the tibia length ratio.

### 2.4. Plasma Determinations

Blood samples were collected in heparinized tubes, cooled in ice and centrifuged for 10 min at 3500 rpm at 4 °C and the plasma frozen at −80 °C. Plasma levels of lipopolysaccharide (LPS) was determined with the Limulus Amebocyte Lyste chromogenic endotoxin quantitation Kit (Lonza, Valais, Switzerland) and plasma L-Lactate levels by L-Lactate Assay kit (Ab65330, Abcam plc, Cambridge, UK) and both kits were used following the manufacturer’s instructions. The enzyme-linked immunosorbent assay kits (IBL International, Hamburg, Germany) to assess plasma noradrenaline were used as instructed by the manufacturer-provided protocol.

### 2.5. Vascular Reactivity Studies

Rings from rat thoracic aorta (3 mm in length) were excised and placed in individual organ chambers with Krebs solution (in mmol L^−1^: NaCl 118, NaHCO_3_ 25, glucose 11, KCl 4.75, CaCl_2_ 2, MgSO_4_ 1.2 and KH_2_PO_4_ 1.2) and infused with carbogen (5% CO_2_ and 95% O_2_) and exposed to a resting tension of 2 g. Isometric tension was recorded using an isometric force-displacement transducer (Letigraph 2000) connected to an acquisition system, as previously described [19].

The concentration–relaxation response curves to acetylcholine (10^−9^–10^−5^ mol/L) were studied in aorta precontracted by phenylephrine (10^−6^ mol/L). In some rings, responses to acetylcholine were determined post-incubation with N^G^-nitro-L-arginine methyl ester (L-NAME, a non-selective competitive inhibitor of nitric oxide synthase, 10^−4^ mol/L) for 30 min. The cumulative concentration-relaxation response curves to nitroprusside (NPS, 10^−11^–10^−7^ mol/L) were performed in the dark in aortic rings without endothelium precontracted by 1 μmol/L phenylephrine. Relaxant responses were expressed relative to precontraction. The presence of functional endothelium was considered by the ability of acetylcholine (1 µmol/L) to induce more than 50% relaxation in vessels maximally precontracted with phenylephrine. Denuded of functional endothelium was considered when no relaxation response to acetylcholine was observed.

### 2.6. Vascular NADPH Oxidase Activity and Measurement of Ex Vivo Vascular Reactive Oxygen Species (ROS)

NADPH oxidase activity was measured in fresh and intact aortic segments by using lucigenin-enhanced chemiluminescence assay, as previously described [21]. Dihydroethidium (DHE, Invitrogen Molecular Probes, Life Technologies S.A. Alcobendas, Madrid, Spain), an oxidative red fluorescent dye served as ROS sensor in optimum cutting temperature (OCT) aortic segments as previously explained [21]. Images were acquired with a fluorescence microscope (Leica DM IRB, Wetzlar, Germany) and quantified with ImageJ (version 1.32j, NIH, http://rsb.info.nih/ij/, accessed date 9 April 2018). Counterstaining with 4,6-diamidino-2-phenylindole 4 dichlorohydrate (DAPI) was done for quantification, allowing comparisons between different sections. As control to determine basal levels, several DOCA group slides were co-incubated with the pan-NADPH oxidases (NOX) inhibitor VAS2870 (10 μmol/L).

### 2.7. Reverse Transcriptase-Polymerase Chain Reaction (RT-PCR) Analysis

As previously described [21] for RT-PCR analysis, total RNA was extracted from the colon and thoracic aortic rings with perivascular adipose tissue homogenizing samples from all groups and retrotranscribing the obtained RNA to cDNA by standard methods. Polymerase chain reaction was performed with a Techne Techgene thermocycler (Techne, Cambridge, UK). mRNA expression was analysed using real time (RT)-PCR. The forward and reverse probes employed are listed in Appendix A.

### 2.8. 16S rDNA V4-V5 Region Sequencing

DNA was extracted from stool samples obtained from of 5–6 animals per group with the quick-DNA fecal/soil microbe kit (Zymoresearch, Irvine, CA, USA). We amplified 16S V4-V5 regions via illumian Miseq v2 2 × 250 bp kit (Illumina, San Diego, CA, USA) with suitable primers [22]. The resulting amplicons were purified using the QIAquick gel extraction kit (QIAGEN, Hilden, Germany) and quantified with Qubit (thermos Fisher Scientific, Waltham, MA). Aliquots from all samples with the same amount of product were pooled together as a library. RT-PCR (Kapa Biosystems, Wilmington, MA, USA) to quantify and study the library prior to Miseq sequencing (Illumina, San Diego, CA, USA). The sequencing data had a Q30 score ≥93.5% and 97.17 ± 0.34% of total cluster passes the filter.

### 2.9. Bioinformatics Analysis

Results from Miseq sequencing were processed with QIIME 1.9.1. Shortly put, reads that included bases with Phred scores lower than 30 were removed and quality-filtered with parameters set as optimized in previous articles [23]. Open reference operational taxonomic unit (OTU)-picking was carried out and taxonomical assignment to the resulting OTUs was done with 97% identity against Greengenes database 13.8. Alpha diversity and unweighted principal coordinate analyses plots using the phylogenic tree-based unifrac distance metric were produced with scripts from QIIME package.

### 2.10. Reagents

The reagents used in this experiment were purchased from Sigma (Alcobendas, Madrid, Spain), unless otherwise specified in the text.

### 2.11. Statistical Analysis

The micro-ecological parameters addressed in this text were quantified using QIIME (PAST 3×). Total reads per sample were used to normalize the reads in each OTU. Only taxa with an expression in our samples > 0.001% were analyzed. Principal component analysis (PCA) was performed to identify significant differences among groups. Only linear discriminant analysis (LDA) scores >2 are shown. Taxonomy was summarized at the genus level within QIIME-1.9.0 and uploaded to the Galaxy platform27 to generate LEfSe/cladogram enrichment plots considering significant enrichment at a *p* < 0.05, LDA score > 2.

Tail SBP evolution and concentration–relaxation responses to acetylcholine were studied with a nested design, with treatment and days and with groups and concentrations, respectively, as fixed factors and the rat as random factor. Results are represented as means ± SEM. If the overall difference was significant, comparisons were performed with Bonferroni’s test and included as an appropriate error. The rest of variables were tested on normal distribution using Shapiro–Wilk normality test and studied with one-way ANOVA and Tukey post hoc test in case of normal distribution, or Mann–Whitney test or Kruskal–Wallis with Dunn’s multiple comparison test in case of abnormal distribution. *p* < 0.05 was considered statistically significant. All the graphs, calculations and statistical analyses were performed using GraphPad Prism version 8.0.0 for Windows, GraphPad Software, Inc., San Diego, CA, USA.

## 3. Results

### 3.1. Doxycycline Treatment Improved Morphologic Variables and Blood Pressure

Basal SBP was 150 ± 2.3 mmHg. Four weeks of DOCA administration and replacement of drinking water with 1% NaCl solution in nephrectomized rats, caused an increase of approximately 30 mmHg in SBP, as compared to control rats, which was maintaining stability for another 4 weeks of vehicle treatment (Figure 1). However, the manipulation of DOCA-salt gut microbiota with the broad-spectrum antibiotic DOX induced a progressive decrease in SBP from the first week, normalizing SBP after 4 weeks of DOX intervention. At the end of the study period, consistent with changes in blood pressure, DOCA-salt-treated rats had significantly increased cardiac, left ventricular and renal weight indices, compared with the animals of the control group. Treatment of DOCA-salt rats with DOX reduced cardiac and left-ventricular hypertrophy, being without effect on renal hypertrophy (Appendix A). Although DOX did not improve renal weight indices, nonetheless, we did not study the DOX effects on DOCA-salt induced renal dysfunction, analyzing parameters as glomerular function or the injury measured by the urine protein expression. It would have been intriguing to have introduced a nephrectomy + DOX group, the lack of this being a limitation of this study.

### 3.2. Doxycycline Treatment Induced Changes in Gut Microbiota in DOCA-Salt Treated Rats

Consistent with other models of hypertension, such as SHR and rats infused with Ang II, we found insignificant differences in bacterial load between control and hypertensive DOCA-salt rats (fecal biomass DNA/fecal amount (ng/mg): 199.0 ± 22.8 and 227.8 ± 28.3, respectively, *p* > 0.05). Doxycycline did not produce any changes in the bacterial load (254.9 ± 17.5 ng/mg). Similar data to those showing that tetracycline minocycline treatment for 4 weeks also did not induce a reduction of bacterial load in Ang II-infused rats [3]. However, Angelakis et al. [24] demonstrated that patients treated with long-term treatment with DOX for 18 months showed significant decrease in total intestinal bacterial content, indicating that the total bacterial concentration significantly decreased with treatment duration.

Fecal DNA was isolated from both control and DOCA-salt animals and 16S ribosomal DNA sequencing were performed to evaluate the gut microbiota composition in both groups. Significant changes in microbial composition were found in DOCA-salt group in comparison with control rats. To further compare the composition of the gut microbiota between control and DOCA-salt rats, three major micro-ecological parameters were determined: (a) Shannon diversity (the combined parameters of richness and evenness), (b) Chao richness (an estimate of a total number of OTUs present in the given community) and (c) Pielou evenness (to show how evenly the individuals in the community are distributed over different OTUs) and number of species. No significant changes between both groups were detectable for diversity and richness; however, Pielou evenness was found to be significantly decreased in DOCA-salt group in comparison to the control rats without any effect of DOX treatment (Figure 2A). Furthermore, DOCA-salt + DOX displayed a modest, but significant lower richness value and a reduction of number of species when compared with DOCA-salt group, pointing to a change in gut bacterial composition due to the administration of the antibiotic, similarly to the pattern previously observed in mice [25,26].

Additionally, we observed that gut microbiota was mainly dominated by Firmicutes and Bacteroidetes phyla, with lower proportion of bacteria belonging to Proteobacteria, Actinobacteria, Tenericutes and Cyanobacteria (Figure 2B and Appendix A). However, we did not find significant differences between the control group and DOCA-salt group in the two main phyla percentages (Figure 2B) and F/B ratio, widely used as a marker of gut dysbiosis (Figure 2C). DOX treatment was unable to induce any changes in gut microbiota composition at phylum level. Reduction in proportion of acetate- and butyrate-producing bacteria has also been used as a marker for dysbiosis in hypertensive animals and patients [3]. However, any differences in the proportion of these short chain fatty acids (SCFA)s-producing bacteria between control and DOCA-salt groups were not observed, as well as the treatment with DOX did not lead to modification of the acetate content but slightly lead to increase of the butyrate content without statistical significance (Figure 2D). Interestingly, we found significant increase of lactate-producing bacteria in feces (Figure 2D) and higher plasma L-lactate levels (Figure 2E) in DOCA-salt rats, which were both abolished by DOX treatment.

Furthermore, it was also found that bacterial taxa (phylum, class, order, family and genus) showed changes in DOCA group as compared to the control rats, according to the LEfSe analysis. In fact, when the relative abundance was represented by LDA considering score >2, 25 taxa increased (as shown in green) and 20 taxa decreased (as shown in red), compared to normotensive animals. The analysis of taxonomy showed a significant increase in the abundance of *Bacilli* class of Firmicutes and more specifically in the order of *Lactobacillales* in the DOCA-salt rats as compared to the control rat group (Appendix A in green), which is confirmed in the afore-mentioned data, describing an increase in lactate-producing bacteria in the DOCA group. Moreover, the relative abundance of 27 taxa was increased (shown in red) in DOCA-treated rats when compared with DOCA-DOX group (Appendix A), concretely an increase of *Bacilli* class and *Lactobacillales* family in DOCA group was observed. Thus, overall DOCA-salt model led to several changes in specific bacteria belonging to Firmicutes phylum, which could be involved in high blood pressure, since they were normalized by DOX treatment.

As DOX treatment caused an alteration in the gut microbial environment, we have also analyzed which genera of bacteria could be involved. Figure 3A shows 3-dimensional scatterplots from principal coordinate analyses visualizing whether the three experimental groups in the input phylogenetic tree have significantly shown different microbial compositions. At genus level, the PCA plot illustrated a separation among all experimental groups (Figure 3A). The main separation was due to DOCA-salt hypertensive group, wherein control and DOX treatment were closer in genera. We found that the gut microbiota of DOCA-salt rats had main higher enrichment of *Lactobacillus* and to a lesser extent to *Sutterella* and *Actinobacteria* and lower percentage of *Oscillospira* than the control group (Figure 3B,C). Previous studies have demonstrated that lactate-producing bacteria (*Lactobacillaceae*, *Bifidobacteriaceae* and *Enterobacteriaceae*) tend to increase in hypertensive rats [3,4,27]. Thus, the treatment with DOX did reduce the accumulation of *Lactobacillus*, which could be involved in causing the decrease in blood pressure in DOCA-salt rats.

### 3.3. Doxycycline Treatment Improved Intestinal Integrity, Colonic Inflammation and Reduced Endotoxemia and Plasma Noradrenaline in DOCA-Salt Rats

The pathogenesis of hypertension is linked to increased gut permeability, inflammation and gut pathology [4]. In this research, we analyzed the plasma level of LPS (an indirect marker of gut permeability) for the investigation of whether the changes observed in gut microbiota composition with hypertension were directly correlated with systemic endotoxemia. As observable in Figure 4A, the plasma endotoxin levels in DOCA-salt animals were higher than those in the control group. Consequently, the treatment with antibiotic DOX significantly showed a decrease in endotoxemia in hypertensive rats. Hence, these results indicate that intestinal permeability may be increasing in this DOCA-salt hypertensive model and allows bacterial components (e.g., LPS) to enter the blood circulation. In this regard, we observed that in DOCA-salt hypertension, the colonic barrier integrity was compromised, since the colonic mRNA expression of tight junction proteins such as occludin, zonula occludens (ZO)-1 and mucins (MUC)-2 and MUC-3 was reduced, as compared to the control rat group. Interestingly, DOX treatment normalized the expression of occludin, ZO-1 and MUC-3 (Figure 4B). Further, the improvement in gut barrier function in hypertensive animal group treated with DOX was highly linked with normalization of the colonic expression of pro-inflammatory cytokines, TNF-α and IL-1β (see Figure 4C). In fact, antibiotic DOX has been previously described to show anti-inflammatory effects through lowering the expression of the genes encoding these pro-inflammatory cytokines [28,29,30]. Santisteban et al. (2017) [4] demonstrated a higher gut sympathetic drive associated with the loss of gut integrity and microbial dysbiosis in hypertensive animals. Furthermore, plasma noradrenaline levels were high in DOCA-salt rats and reduced by DOX treatment (refer to Figure 4D).

### 3.4. Doxycycline Treatment Improved Endothelial Function and Reduced Vascular Oxidative Stress and Inflammation in DOCA-Salt Rats

Vascular endothelial dysfunction is related with enhanced vascular tone and high blood pressure [31]. Aortas from DOCA-salt treated animals presented a decrease in endothelium-dependent relaxant responses to acetylcholine compared to the control group (Emax = 41.4 ± 4.2% vs. 88.2 ± 3.6%, respectively, *p* < 0.01). In contrast, the treatment of DOCA-salt rats with DOX, improved the endothelium-dependent relaxation response to acetylcholine (Emax: 60.5 ± 3.4%, *p* < 0.01 vs. DOCA-salt group) (refer to Figure 5A). The relaxant responses in all experimental groups were suppressed after incubating with L-NAME (Appendix A), which shows that in these vessels, acetylcholine-induced relaxation was almost entirely dependent on endothelium-derived NO. In addition, the endothelium-independent relaxant responses to NO donor-nitroprusside were unchanged in DOCA-salt rats and also after treatment with DOX in comparison to the control group (Appendix A), a clear indication that the impaired response to endothelium-derived NO was not due to a defect in NO signaling in vascular smooth muscle.

NADPH oxidase ROS synthesis is highly relevant in endothelial dysfunction in DOCA-salt hypertension. In this experiment, the selective NADPH oxidase inhibitor VAS2870 increased the aortic relaxant capacity to acetylcholine in DOCA-salt mice displaying similar relaxations to those shown in the control group (data not shown). Furthermore, vascular ROS levels were subjected to close study through ethidium red fluorescence in sections of aorta incubated with DHE. Positive red nuclei were observed in endothelial, medial and adventitial cells from sections of aorta incubated with DHE (Figure 5B). As expected, rings from DOCA group displayed significant staining in endothelial, medial and adventitial cells, which was higher than in control rats. These effects were prevented by DOX administration (Figure 5B). Since NADPH oxidase is the major source of ROS in the vascular wall, we found that the presence of the pan-NOX inhibitor VAS2870 reduced the staining of the aorta sections from DOCA rats (Figure 5B). In fact, NADPH oxidase activity was higher in the aortic rings of DOCA-salt group than in aortic rings of control group (Figure 5C). Hence, this was accompanied with a significant mRNA increase in the catalytic NOX-4 and regulatory, p22^phox^ and p47^phox^, subunits in DOCA-salt rats (Figure 5D). Chronic administration of DOX significantly reduced the increase of NADPH oxidase activity and the gene overexpression of NOX subunits in DOCA-salt group, respectively (Figure 5C,D).

Bacterial endotoxin LPS has been described to activate Toll-like receptor (TLR)-4 in the vasculature, which induces pro-inflammatory cytokines in the vascular walls [32,33] and increases NADPH oxidase-dependent ROS synthesis, mainly superoxide, which can react with and inactivate NO in the vascular system [31,34]. In fact, we found that the mRNA expression of TLR4 in homogenates of thoracic aortic rings was higher in aorta from DOCA-salt group as compared to the control rat group. In correlation with the lower plasma endotoxin levels in DOCA-DOX group, we also observed that up-regulation of TLR4 was abolished by DOX treatment (Figure 5D). Likewise, pro-inflammatory cytokines TNF-α, IL-1β and IL-6 in aorta were higher in DOCA group than in control group and DOX antibiotic treatment significantly reduced mRNA levels of, IL-1β and IL-6 without changes in TNF-α gene expression (Appendix A).

It has been described that gut microbiota is an important factor involved in the control of blood pressure, as a consequence of its effect in T-cell activation in gut immune system and vascular T-cells accumulation [21]. We observed that CD4 mRNA, marker of T cells, was higher in aortic homogenates from DOCA-salt group as compared to the control group (Figure 6A) and was significantly reduced by DOX treatment. Furthermore, aortic infiltration of pro-inflammatory cells, such as T helper 17 (Th17, Figure 6B), measured by the mRNA expression levels of both retinoid-related orphan receptor-γ (RORγ) and IL-17 and Th1 (T-bet, Figure 6C), analyzed by the mRNA expression of both T-bet and IFN-γ, were also higher in DOCA-salt animals as compared to the control rat group, but were unchanged by DOX treatment. However, the mRNA levels of anti-inflammatory cells, such as regulatory T cells (Treg) measured by mRNA expression of fork-head box P3 (FoxP3) and IL-10 were lower in aortic homogenates from DOCA-salt, than in control group and DOX treatment significantly suppressed these changes (Figure 6D).

## 4. Discussion

The most significant finding of this study is that the tetracycline antibiotic DOX treatment significantly contributes to the improvement of the vascular dysfunction and to a decrease in blood pressure in a low renin model of hypertension. These DOX effects are due to gut microbiota which initiates underlying mechanism and non-antimicrobial effects that contributes to an increase in NO bioavailability. In fact, the study provides evidence for the gut microbiota involvement and broad-spectrum antibiotic treatment as a strategic therapy for blood pressure regulation in salt-sensitive low renin hypertension. The key aspects of these findings are as follows: (1) chronic treatment with the tetracycline DOX prevents the increase of blood pressure and also reduces cardiac and left-ventricular hypertrophy induced by DOCA-salt administration; (2) DOX treatment decreases both increased lactate-producing bacteria, mainly *Lactobacillus* and high plasma L-lactate; (3) broad-spectrum antibiotic treatment also improves colonic integrity and normalizes endotoxemia; (4) DOX improves the endothelial function due to a diminished NADPH oxidase-dependent ROS production; and (5) finally, the antibiotic DOX treatment increases Treg infiltration and IL-10 in the vascular wall in mineralocorticoid excess-treated rats.

In recent years, it has been shown that gut microbiota plays a major role in blood pressure control [22]. DOCA-salt model incorporates a high-salt diet, as well as producing neurogenic and a low-renin form of hypertension, similarly to what is described in the human population [18,35]. Previous studies have indicated that broad-spectrum antibiotics can modify the gut microbiota and have been used strategically for modulating the gut microbiota and in reduction of blood pressure in animal models of hypertension and in patients with resistant hypertension [3,36,37]. The tetracycline antibiotic minocycline can restore gut microbiota and attenuate the high blood pressure induced by chronic Ang II infusion [3]. DOX also belongs to the same antibiotic group with the limited ability to go through the blood–brain barrier as demonstrated by several experimental studies that attenuates the increase in SBP in different animal models of hypertension [12,13,14,15]. In this experiment, we also found antihypertensive effects of DOX in DOCA-salt rats, independently of a direct effect in neuroinflammation. However, the increased sympathetic activity in hypertensive rats, measured by plasma noradrenaline concentration, was reduced by DOX treatment, suggesting a gut–neuronal communication, as demonstrated by reduced sympathetic drive after fecal microbiota transplantation from normotensive to hypertensive rats [38,39].

We observed that in DOCA-salt animals, the gut microbiota was characterized by a reduced evenness, with no significant changes in F/B ratio and a higher proportion of lactate-producing bacteria, which was accompanied with high levels in plasma lactate. This signature of gut microbiota is consistent with the previous data shown in mice and rats with DOCA-salt hypertension [7,40,41]. The manipulation of the gut microbiota with oral antibiotic DOX reduced richness and number of species; however, it did not alter the bacterial load. This is not surprising, because minocycline treatment of hypertensive rats did not result in any reduction of bacterial load [3], which required a combination of at least of three antibiotics. DOX treatment decreased lactate-producing bacteria, which also may be correlated with the reduction of plasma lactate levels that could contribute to the beneficial effect to reduce the BP; it has already been detailed that plasma lactate level is linked to increased blood pressure [42,43]. In fact, the decrease in gut lactate-producing bacterial populations by the immunosuppressive drug moletil mycophenolate and consumption of a high-fiber diet with *Bifidobacterium breve* reduces plasma lactate and hence prevents the rise of blood pressure, attenuation of both renal and cardiac fibrosis and the improvement of endothelial function in DOCA-salt rats [7,40,41].

In DOCA-salt rats, a leaky intestinal barrier has been previously described [40,41], characterized by increased permeability, combined with altered tight junction proteins including an increase in plasma LPS levels. DOX has been shown to produce beneficial effects on the intestinal epithelium, as well as the inhibition of inflammation and immunomodulatory effects, in addition to its essential antibiotic effect [44,45,46]. In support of these data, we also found that DOX produces an improvement in barrier function by an increase in colonic mRNA levels of tight junction protein occludin and ZO-1 and MUC-3. A limitation of this study is the lack of protein expression analysis. In an in vitro intestinal ischemia/reperfusion injury model, DOX has shown antiapoptotic properties attenuating of hypoxia-induced cell damage by reducing expression of caspase-3 in human intestinal CaCo-2 cells [47], which may also contribute to the repair of the disruption of gut barrier integrity in DOCA-salt rats. Worth noting is its capacity to influence the expression of makers of mucosal barrier function. In this regard, we found that DOX can highly inhibit the expression of colonic pro- inflammatory cytokines TNF-α and IL-1β. In conjunction, these results suggest that DOX can help in restoring intestinal mucosal barrier functionality and preventing the access of LPS into the blood stream, one of the main factors that promote hypertension through alterations to the vascular endothelium [48].

Established evidence from animal and in vitro studies supports the pathological role of LPS in hypertension through the activation of TLR4 receptor and, hence, the resulting activation of NADPH oxidase and an inducement of production of series of pro-inflammatory cytokines in vascular endothelium [49]. Thus, we found higher plasma levels of LPS and aortic increased TLR4 mRNA levels in DOCA-salt than control rats, similarly shown by [40]. Thus, the gut translocating LPS may contribute (at least partially) to the inducement of endothelial dysfunction in DOCA rats. Previous studies have demonstrated that tetracycline DOX regulates the expression of TLRs and cytokines in septic conditions [50,51]. In rats, antibody blockade of TLR4 lowers blood pressure in SHR and DOCA salt rats, suggesting that TLR4 plays an important role in blood pressure regulation. Furthermore, we observed that DOX treatment consistently reduced endotoxemia in DOCA-salt animals, while protecting endothelium-dependent vasorelaxation through down-regulation of TLR4 mRNA, reducing NADPH oxidase activity and ROS levels. These inhibitory effects of DOX on the TLR4/LPS-signaling pathway, including the reduction of the pro-inflammatory cytokines IL-1β and IL-6, could explain the improvement of endothelial dysfunction and diminishing of blood pressure.

The tetracycline DOX regulates the immune response in both presence and absence of LPS through modulation of Toll-like receptors and cytokine production (IL-1β and IL-6) [52,53]. Additionally, there are accumulative evidence indicating that Th17, Treg and Th17/Treg imbalance are involved in the initiation and the development of increased blood pressure in salt-sensitive hypertension [54,55]. Furthermore, the pro-inflammatory cytokine IL-17, released in part by Th17, induces ROS-mediated endothelial dysfunction [56], whereas anti-inflammatory cytokine IL-10, produced by Tregs, is shown to protect endothelial function in hypertension by attenuates NADPH oxidase activity [57]. In this study, we also found higher Th17 and lower Tregs in aorta from DOCA-salt animals and the treatment with DOX increased the numbers of Treg cells as well as the levels of the anti-inflammatory IL-10 in DOCA salt rats. This could suggest that these effects could be associated with improved endothelial function and reduced arterial blood pressure. In fact, it has been demonstrated that Treg transfer is able to: (1) prevent increased blood pressure, (2) decreased vasodilation and (3) vascular remodeling of resistance arteries in DOCA-salt-induced hypertension [58]. These changes have been related to the IL-10 released from Tregs, which attenuates NADPH oxidase activity and superoxide production in aorta, kidney and heart after Treg treatment [58,59]. Accordingly, we discerned that the selective inhibitor of NADPH oxidase -VAS2879 improved the impaired relaxation to acetylcholine in aorta from DOCA-salt group. DOX treatment was able to increase the cytokine IL-10 by Tregs in DOCA salt rats, which may be correlated to reduced vascular up-regulation of NADPH oxidase subunits and its activity in addition to an improvement of NO-dependent relaxations. Furthermore, in vivo and in vitro studies have also shown that DOX may have significant ROS scavenging properties [13]. All these results suggest that the reduction of ROS levels in the vascular wall and the subsequent prevention of NO inactivation, constitutes a pivotal mechanism, involved in the DOX protective effects on endothelial function and the decrease of SBP in DOCA-salt hypertension. We did not, however, test if there is a mechanistic link between high vascular IL-10 level and both transcriptional and function changes found in DOCA-salt in the vascular wall.

## 5. Conclusions

Taken together, the results presented in this study demonstrate that antibiotic treatment with DOX in DOCA-salt hypertension aid the improvement of endothelium-dependent relaxation and significantly reduces blood pressure. These protective effects seem to be associated, at least in part, with a decrease in the vascular oxidative stress; by restoring the expression of NOX subunits to normal values resulting in an improved gut barrier function and lower plasma levels of lactate, LPS and vascular proinflammatory cytokines. In addition, the increased vascular Tregs accumulation contributed to the vascular protective effects of DOX. Thus, in this study, we noted that tetracycline DOX through direct effects on gut microbiota and its anti-inflammatory and immunomodulatory properties may help reduce the vascular dysfunction associated with DOCA-salt hypertension. We therefore concluded that understanding the impact of broad-spectrum antibiotics on gut microbiota and knowing its non-microbial effects on gut barrier and vascular wall can significantly help guide appropriate treatment strategies for patients with low renin hypertension.

## Figures and Tables

**Figure 1 nutrients-13-02971-f001:**
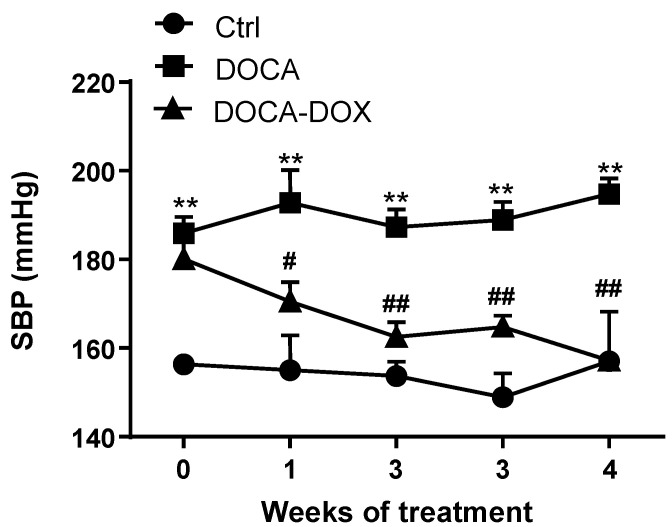
Doxycycline (DOX) prevented the rise in blood pressure in DOCA-salt rats. Time course of systolic blood pressure (SBP) assessed with tail-cuff plethysmography. Results represented as mean ± SEM (*n* = 8). ** *p* < 0.01 versus Control (Ctrl) group. ^#^
*p* < 0.05 and ^##^
*p* < 0.01 versus the nontreated DOCA group.

**Figure 2 nutrients-13-02971-f002:**
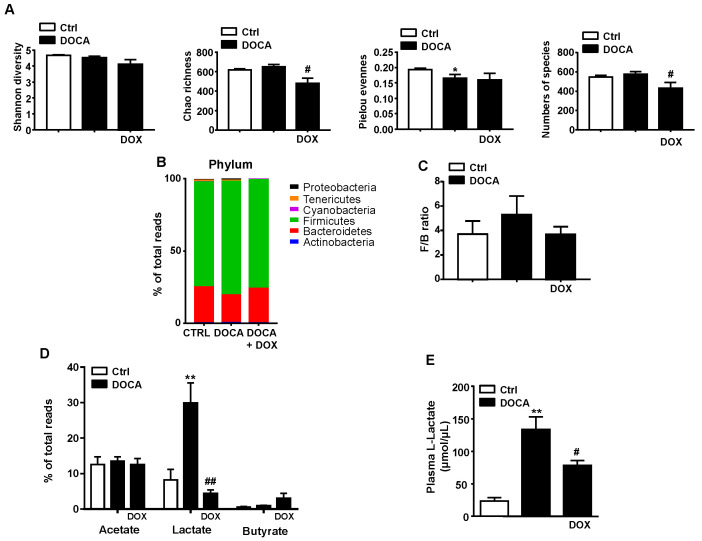
Effects of Doxycycline (DOX) on phyla changes in the micro-ecological parameters and gut microbiota composition. Analysis of the 16S ribosomal DNA in stool samples. (**A**) Bacterial diversity, richness, evenness and numbers of species as markers for differences in microbial composition among groups. (**B**) Phylum breakdown of the most abundant bacterial communities in each group. (**C**) The Firmicutes/Bacteroidetes (F/B) ratio as a biomarker of gut dysbiosis. (**D**) Relative proportions of acetate-, lactate- and butyrate-producing bacteria. (**E**) Plasma lactate concentrations (µmol/µL) in control (Ctrl) and DOCA-salt rats. Data represented as means ± sem (*n* = 6–8). * *p* < 0.05 and ** *p* < 0.01 versus Ctrl group; ^#^
*p* < 0.05, ^##^
*p* < 0.01 versus DOCA-salt group.

**Figure 3 nutrients-13-02971-f003:**
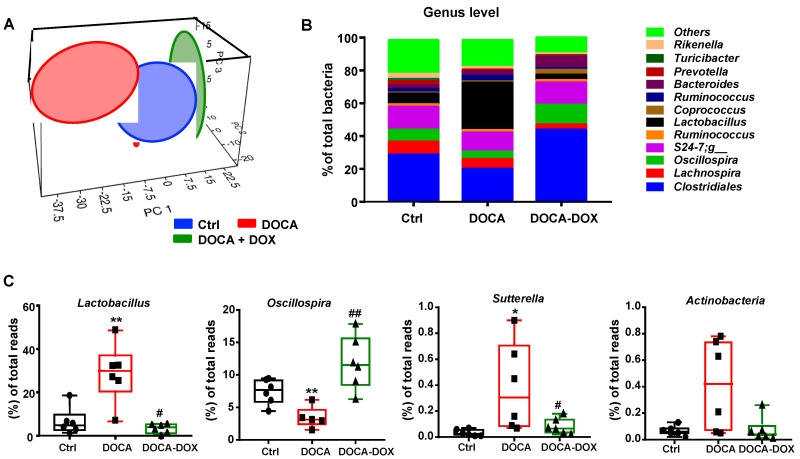
Effects of Doxycycline (DOX) on genera shifts in gut microbiota composition. (**A**) Principal components analysis for gut microbiota. (**B**) Main bacterial genera as percentages of total bacteria. DOXI resulted in distinct populations of bacteria at the genus level. (**C**) Bacterial genera *Lactobacillus, Osciillospira, Suterella and Actinobacteria* in the gut microbiota in control (Ctrl) and DOCA-salt rats. Data represented as means ± SEM (*n* = 6). * *p* < 0.05, ** *p* < 0.01 versus the Ctrl group; ^#^
*p* < 0.05, ^##^
*p* < 0.01 versus the untreated DOCA-salt group.

**Figure 4 nutrients-13-02971-f004:**
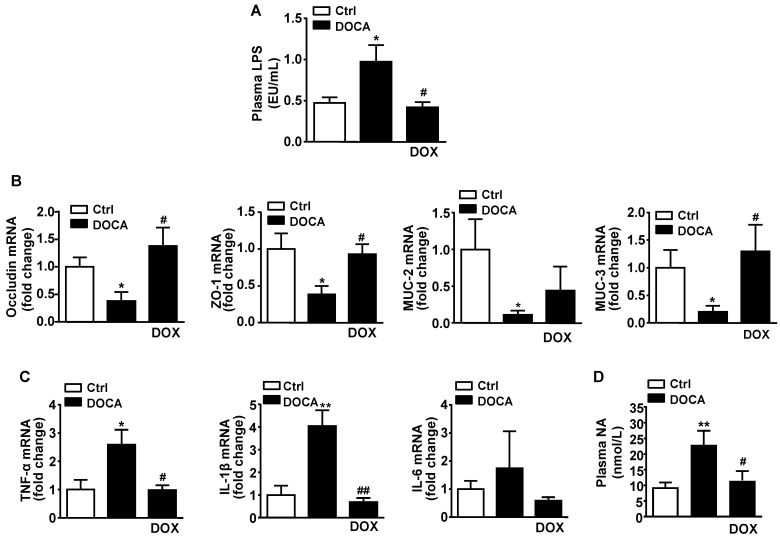
Doxycycline (DOX) improved intestinal integrity and inflammation, ane reduced endotoxemia and plasma noradrenaline levels in DOCA-salt rats. (**A**) Plasma endotoxin concentrations (EU/mL, endotoxin units/mL). (**B**) mRNA levels in colon of occludin, zonula occludens-1 (ZO-1), mucin (MUC)-2 and MUC-3. (**C**) mRNA levels of pro-inflammatory cytokines in colon, tumour necrosis factor-α (TNF-α) and interleukin (IL)-6. (**D**) Plasma noradrenaline (NA) levels (nmol/L). Data represented as mean ± SEM (*n* = 8). * *p* < 0.05 and ** *p* < 0.01 versus Control (Ctrl) group. ^#^
*p* <0.05 and ^##^
*p* < 0.01 versus the non-treated DOCA group.

**Figure 5 nutrients-13-02971-f005:**
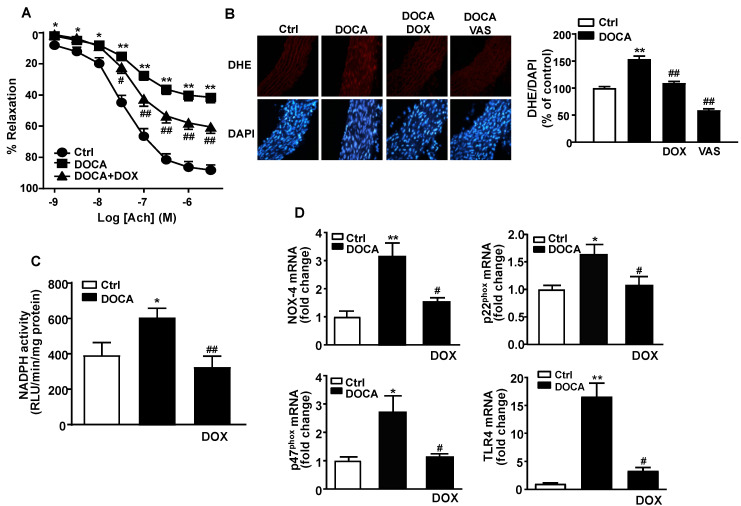
Effects of Doxycycline (DOX) on endothelial function, vascular oxidative stress and NOX pathway. (**A**) Endothelial reactivity to acetylcholine (Ach) in aortas precontracted with phenylephrine in control (Ctrl) and DOCA-salt groups. (**B**) Top images display arteries incubated with dihydroethidium (DHE), which produces a red fluorescence when oxidized to ethidium by ROS. Bottom images depict the blue fluorescence of the nuclear dye DAPI (original magnification, ×400). Averaged values (*n* = 8 rings from different rats) of the red ethidium fluorescence normalized to the blue DAPI fluorescence. (**C**) NADPH oxidase activity measured by lucigenin-ECL and (**D**) Expression of NADPH oxidase subunits NOX-4 and p22^phox^ and Toll-like receptor (TLR)4 at the level of mRNA by RT_PCR in Ctrl and DOCA-salt rats. Data are represented as means ± SEM (*n* = 8). * *p* < 0.05, ** *p* < 0.01 versus Ctrl group; ^#^
*p* < 0.05, ^##^
*p* < 0.01 versus nontreated DOCA group.

**Figure 6 nutrients-13-02971-f006:**
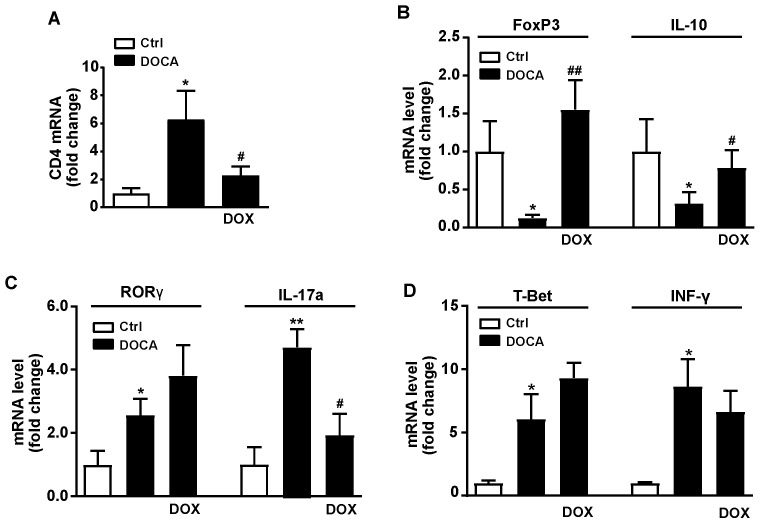
Effects of Doxycycline (DOX) on aortic T-cells infiltration. (**A**) Th (CD4) cells infiltration in aorta. (**B**) Regulatory T cells (Treg) infiltration in aorta quantified by proxy with FoxP3 and interleukin (IL)-10 mRNA levels. (**C**) Aortic T helper (Th)17 infiltration determined with mRNA levels of RORγ and IL-17a. (**D**) Aortic Th1 (T-bet) infiltration and IFN-γ in control (Ctrl) and DOCA-salt rats. Data are represented as means ± sem (*n* = 8). * *p* < 0.05, ** *p* < 0.01 versus Ctrl group; ^#^
*p* < 0.05, ^##^
*p* < 0.01 versus DOCA-salt group.

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
