# Peer review of "Changes in Gut Microbiota Induced by Doxycycline Influence in Vascular Function and Development of Hypertension in DOCA-Salt Rats"

_nutrients, 2021, doi:10.3390/nu13092971_

Round 1

Reviewer 1 Report

The research topics are "Changes in gut microbiota induced by doxycycline influence in vascular function and development of hypertension in DOCA-salt rats ", which is interest research and suitable for the trends, this manuscript doxycycline treatment reduced systolic blood pressure, improved endothelial dysfunction, and reduced oxidative stress and inflammation in aorta. Doxycycline decreased lactate-producing bacterial population and plasma lactate levels, improved gut barrier integrity, normalized endotoxemia, plasma noradrenaline levels, and restored the Treg content in aorta. It is recommended to accept after minor revision.

1.    Line 118: please explain what is 18-hour animals.
2.    Line 177: please explain there are 8 rats in each group, why only choose 5-6?
3.    Line 241-244: Please add the reference (DOI: 10.1128/AAC.02437-14) in the discussion. The author should explain the effects of DOX on the intestinal flora and the total number of bacteria based on the literature on DOX treatment.
4.    Line 269-273: The article in this paragraph does not match the content of the figure. For example, the use of SCFA description is obviously inconsistent with the butyric acid in Fig. 2D.
5.    There seems to be an error in the statistical labeling in Fig 4, such as Fig 4B, ZO-1's *, and ##, please check again.

Author Response

Reviewer: 1

We thank the reviewer for the helpful comments and the positive criticisms. Following his/her suggestions we have made changes to the text, which, we believe, have improved the manuscript.

R#1.1. Line 118: please explain what is 18-hour animals.

R#1.1. Answer. It does not make any sense since the word fasting is missing. It has been modified in the manuscript “18 hours fasting animals”.

R#1.2.  Line 177: please explain there are 8 rats in each group, why only choose 5-6?

R#1.2. Answer. Obtaining statistically significant results with the sample size 5-6/group was based on the Minimum and Maximum Sample Sizes for ANOVA Design (doi: 10.21315/mjms2017.24.5.11). Therefore adding 2-3 more samples would not have led to any increase in the chance of getting significant results.

R#1.3. Line 241-244: Please add the reference (DOI: 10.1128/AAC.02437-14) in the discussion. The author should explain the effects of DOX on the intestinal flora and the total number of bacteria based on the literature on DOX treatment.

R#1.3. Answer. This is an interesting suggestion. We have included in the line 244 the following “Similar data to those showing that tetracycline minocycline treatment for 4 weeks also did not induce a reduction of bacterial load in Ang II-infused rats [3]. However, Angelakis et al [24] demonstrated that patients treated with long-term treatment with DOX for 18 months showed significant decrease in total intestinal bacterial content, indicating that the total bacterial concentration significantly decreased with treatment duration.”

Angelakis E, Million M, Kankoe S, Lagier JC, Armougom F, Giorgi R, Raoult D. Abnormal weight gain and gut microbiota modifications are side effects of long-term doxycycline and hydroxychloroquine treatment. Antimicrob Agents Chemother. 2014 Jun;58(6):3342-7. doi: 10.1128/AAC.02437-14. Epub 2014 Mar 31. PMID: 24687497; PMCID: PMC4068504.

R#1.4. Line 269-273: The article in this paragraph does not match the content of the figure. For example, the use of SCFA description is obviously inconsistent with the butyric acid in Fig. 2D.

R#1.4. Answer. For a better understanding between the text and the figure we have added the following in the line 273 “as well as the treatment with DOX did not lead to modification of the acetate content but slightly lead to increase of the butyrate content without statistical significance”.

R#1.5.  There seems to be an error in the statistical labeling in Fig 4, such as Fig 4B, ZO-1's *, and ##, please check again.

R#1.5.  Answer. We have checked the statistical results and we agree with the reviewer, and it should appear just one # so it has been replaced in the graph.

Reviewer 2 Report

In this paper, Robles-Vera and Colleagues explore the effect of doxycycline on microbiome, vascular function and hypertension in a rat model of DOCA-salt-induced hypertension.

The paper is interesting but a few comments may be addressed to it.

  • The manuscript needs a substantial English language revision as there are several grammar mistakes.
  • The Authors should be commended for the number of molecules they analyzed, however, a few western blot images could enhance the overall importance of their findings.
  • I would suggest including in the manuscript the limitations of this work.
  • I think that, in the limitations of the study, the lack of a control nephrectomy+DOX group should be acknowledged.
  • Please define acronyms before their first use in the text (as an example, page 2, B and B/F ratio; page 4, ROS and OCT etc.).
  • “Rats were randomly divided into three experimental groups: in normotensive rats: (I) control (uninephrectomized, n = 8); and two in DOCA-salt hypertensive rats: (II) DOCA-salt (uninephrectomized, and with free access to drinking water containing 1% NaCl, n = 8), (III) Doxycycline (DOX)-treated, DOCA-salt + DOX (uninephrectomized, and 1% NaCl in drinking water and DOX 25 mg kg−1 by oral gavage mixed in 1 ml of 1% methylcellulose, once daily, n = 8).” This sentence is not very clear, please rephrase.
  • Although the Authors refer to a previous paper for a detailed description, please explain better in the Methods the animal model, in particular the DOCA treatment, as there is no mention of the dose and administration schedule at all.
  • Please add the manufacturer for the VAS2870 inhibitor, is it always Sigma?
  • Please add, in the Statistics, the program used to perform them. Is it Graphpad Prism?
  • I would suggest including in figure 1 the SBP values before the treatment start.
  • Discussion, line 448 to 464, is a repetition of something already stated in the introduction. Since the paper is already quite long, I would suggest shortening or even removing it.

Author Response

Reviewer 2

We thank the reviewer for the helpful comments and the positive criticisms. Following his/her suggestions we have made changes to the text, which, we believe, have improved the manuscript.

R#2.1. The manuscript needs a substantial English language revision, as there are several grammar mistakes.

R#2.1. Answer. Following the suggestion of the reviewer the authors have reviewed the whole manuscript and corrected the grammatical errors.

R#2.2. The Authors should be commended for the number of molecules they analysed, however, a few western blot images could enhance the overall importance of their findings.

Answer. We agree with the referee about the relevance of quantification of protein expressions of some molecules. Determining protein expression requires an account of sample which regrettably we did not have. None the less we are aware that this is a limitation of the study (pag 12, last line).

R#2.3. I would suggest including in the manuscript the limitations of this work. I think that, in the limitations of the study, the lack of a control nephrectomy+DOX group should be acknowledged.

R#2.3. Answer. Following your suggestion, we have included in the results section, line 232 “Although DOX did not improve renal weight indices, none the less we did not study the DOX effects on DOCA-salt induced renal dysfunction, analysing parameters as glomerular function or the injury measured by the urine protein expression. It would have been intriguing to have introduced a nephrectomy+DOX group, being a limitation of this study.”

R#2.4. Please define acronyms before their first use in the text (as an example, page 2, B and B/F ratio; page 4, ROS and OCT etc.).

R#2.4. Answer. Following your suggestion, we have checked the whole text defining acronyms before their first time.

R#2.5. “Rats were randomly divided into three experimental groups: in normotensive rats: (I) control (uninephrectomized, n = 8); and two in DOCA-salt hypertensive rats: (II) DOCA-salt (uninephrectomized, and with free access to drinking water containing 1% NaCl, n = 8), (III) Doxycycline (DOX)-treated, DOCA-salt + DOX (uninephrectomized, and 1% NaCl in drinking water and DOX 25 mg kg-1 by oral gavage mixed in 1 ml of 1% methylcellulose, once daily, n = 8).” This sentence is not very clear, please rephrase.

R#2.5. Answer. Following the suggestion of the reviewer we have rephrased the sentence “Rats were randomly divided into three experimental groups (n = 8): (I) control, (II) DOCA-salt (1% NaCl in drinking water) and (III) DOX-treated DOCA-salt (1% NaCl in drinking water and DOX 25 mg kg-1 by oral gavage mixed in 1 ml of 1% methylcellulose, once daily)”.

R#2.6. Although the Authors refer to a previous paper for a detailed description, please explain better in the Methods the animal model, in particular the DOCA treatment, as there is no mention of the dose and administration schedule at all.

R#2.6. Answer. Following your suggestion, we included this new information in the methods section line 104 “DOCA-salt hypertension model was induced in uninephrectomized animals by intramuscular injection of DOCA 12.5 mg/0.5 mL/rat per week for four consecutive weeks as previously is described in detail [19]”.

R#2.7. Please add the manufacturer for the VAS2870 inhibitor, is it always Sigma?

R#2.7. Answer. VAS2870 inhibitor, as many of the compounds we have used in this study, was obtained from Sigma.  

R#2.8. Please add, in the Statistics, the program used to perform them. Is it Graphpad Prism?

R#2.8. Answer. Following your suggestion, we included this new information in the Statistical analysis section “All the graphs, calculations, and statistical analyses were performed using GraphPad Prism version 8.0.0 for Windows, GraphPad Software, Inc., San Diego, CA”.

R#2.9. I would suggest including in figure 1 the SBP values before the treatment start.

R#2.9. Answer. In the figure 1, at week 0 which is represented by SBP values before the commencement of the DOX-treatment. For clarity purpose we have included in the results section, the value of basal SBP (as referenced in red) of group of animals that were treated with DOCA for four weeks before the DOX treatment was administered.

R#2.9. Discussion, line 448 to 464, is a repetition of something already stated in the introduction. Since the paper is already quite long, I would suggest shortening or even removing it.

R#2.9. Answer. Following your suggestion, we have reduced the content of the paragraph.

In recent years, it has been shown that gut microbiota plays a role in blood pressure control [22]. DOCA-salt model incorporates a high-salt diet but also produces neurogenic and a low-renin form of hypertension, similar to what is described in the human population [18, 35]. Several intervention studies have showed that blood pressure levels, in different animal models for hypertension, can be modified by fecal microbiota transplantation and antibiotic treatment [3]. Previous studies have indicated that broad-spectrum antibiotics can modify the gut microbiota and have been used as strategy for modulating the gut microbiota and to reduce blood pressure in animal models of hypertension and in patients with resistant hypertension [3, 36, 37]. The tetracycline antibiotic minocycline is able in restoring gut microbiota and attenuate the high blood pressure induced by chronic Ang II infusion [3]. Additionally, minocycline is an anti-inflammatory antibiotic that penetrates through the blood-brain barrier, showing evidence that there may be confounding neurological variables acting in concert with the gut microbiota to prompt the disparate BP effects of this antibiotic [38]. DOX belongs also to the same antibiotic group with the limited ability to go through the blood-brain barrier and several experimental studies have demonstrated that attenuates the increase in SBP in different animal models of hypertension [12, 13, 14, 15].

Round 2

Reviewer 2 Report

The Authors satisfactory addressed my comments. I think the manuscript still requires an extensive language revision by a native English speaker, especially the newly added sentences. Moreover I would suggest moving the limitations of the study to the end of the Discussion and not acknoeledging them in the Results.